# Does Technological Innovation Curb O_3_ Pollution? Evidence from Three Major Regions in China

**DOI:** 10.3390/ijerph19137743

**Published:** 2022-06-24

**Authors:** Wen-jun Wang, Yan-ni Liu, Xin-ru Ying

**Affiliations:** School of International Business, Shaanxi Normal University, Xi’an 710119, China; wwjunxida@snnu.edu.cn (W.-j.W.); yingxinru@snnu.edu.cn (X.-r.Y.)

**Keywords:** dynamic spatial Dubin model, technological innovation, O_3_ pollution, inter-regional differences

## Abstract

At the end of 2020, when China’s three-year Blue Sky Protection Campaign was successfully concluded, the main pollutants, led by O_3_, increased instead of decreasing, creating a new air pollution problem. In this paper, the impact of the technological innovation level on O_3_ pollution and its inter-regional differences across three major regions from 2014 to 2019 are studied using the dynamic spatial Durbin model. Generally, in terms of ozone pollution showing significant spatial correlation, technological innovations in China are still not effective in curbing ozone pollution. Furthermore, technological innovation is a key factor affecting ozone pollution, and it is heterogeneous, demonstrating that the impact of technological innovation on O_3_ pollution is different among regions. Technological innovation in Beijing–Tianjin–Hebei significantly reduces local O_3_ pollution with spillover, while technological innovation in the Yangtze River Delta instead significantly exacerbates local O_3_ pollution, and the impact of technological innovation on O_3_ pollution in the Fenwei Plain is not significant. Third, other factors in O_3_ pollution also differ between regions, with the number of cars and the amount of foreign capital actually utilized being the main factors. Therefore, we should pay attention to the spillover of O_3_ pollution and technological innovation and strengthen regional cooperation according to our own characteristics to effectively suppress O_3_ pollution. Finally, the findings of this paper are representative, which provides a possible reference for other similar national or regional studies.

## 1. Introduction

Ozone (O_3_) is produced rapidly in polluted air via the photochemical oxidation of volatile organic compounds (VOCs) in the presence of nitrogen oxides (NOx ≡ NO + NO_2_). VOCs originate from both anthropogenic and biogenic sources; NOx mainly comes from fuel combustion [1]. Exposures to ambient ozone can contribute to risks of respiratory or circulatory mortality and also induce plant cell death and yield reductions [2,3]. O_3_ pollution in China has become progressively more severe since 2013. All ozone metrics averaged from Chinese urban sites have increased significantly since 2013, and surface ozone levels in China were significantly higher than those in other regions reported in TOAR [4,5]. As for Beijing–Tianjin–Hebei and the Yangtze River Delta, their ozone concentrations exceed the ambient air quality standard by 100–200% [6]. Moreover, the total premature respiratory mortalities attributable to ambient MDA8 ozone exposure in 69 Chinese cities were 64,370 in 2019, an increase of 60% compared to 2013 levels [4]. While the concentrations of other major pollutants are on a general decreasing trend, O_3_ concentrations are increasing instead of decreasing, generating a new air pollution problem in China and requiring urgent action.

Technological innovation is a key factor that cannot be ignored in the study of atmospheric problems such as ozone pollution. On the one hand, the application of new technologies may have accelerated regional ozone pollution, such as the oil and gas production expansion event in the United States [7]; on the other hand, it may also have mitigated ozone pollution hazards to human health [8] and crops [9] by improving outdated production practices. Related studies on O_3_ pollution in China have focused on chemical precursors [10,11], meteorological conditions [12,13], spatial and temporal variations [14,15], regional transport [10], sources [16,17], and harmful effects on humans and crops [18] from geographic, chemical, and biological dimensions. However, fewer studies have examined the effect of technological innovation on O_3_ pollution in the Chinese region, and when the focus is zoomed in on the effect of technological innovation on environmental pollution, the conclusions of a large number of studies are as shown below.

For the relationship between technological innovation and environmental pollution, the existing literature uses the number of patent applications [19,20,21], the number of patents granted [22], and R&D investment [23] as variables to measure technological innovation to study its impact on environmental pollution. However, it is controversial as to whether technological innovation can improve environmental pollution, and there are three main views. First, technological innovation improves environmental pollution: the “technology effect” in the EKC mechanism posits that in the process of a country’s economic growth, technological progress will improve productivity and resource use efficiency and reduce factor inputs in the production process, thus weakening the impact of production on the natural environment; meanwhile, the development, use, and upgrading of clean technologies will also effectively reduce pollution emissions [24]. A large number of studies have argued the role of technological innovation based on the EKC curve, verifying that technological progress is a necessary condition for the inflection point of the EKC, which is an inevitable way to solve the environmental pollution problem [25,26]. The improvement effect of technological innovation is even greater than the effect of improving energy use efficiency, adjusting the industrial structure, and strengthening investment in pollution control [27,28]. Second, technological innovation exacerbates environmental pollution: technological innovation contributes significantly to economic growth but is also a source of environmental degradation [29,30]. Technological innovation is more about the pursuit of production efficiency, not the impact on environmental pollution. The increasing scale of economic development will consume a large amount of scarce and non-renewable resources, and technological innovation may form new sources of pollution while increasing productivity, thus limiting the effect of technology on the EKC. Even if there is environmental improvement, it is a temporary phenomenon due to technological limitations [31,32,33,34]. Third, there are uncertainties in the impact of technological and technological innovation on environmental pollution, including no impact [35,36] and co-existing positive and negative relationships, showing an N-shaped relationship [37], an inverted U-shaped relationship [38,39], and a complex association with intellectual property protection intensity as a threshold nonlinear relationship [22]. In addition, the direction and degree of influence of technological innovation on environmental pollution vary among cities, which are related to factors such as the stage of economic development, city size, and the level of urban technological innovation in the study area [40,41].

To sum up, the effects of technological innovation on environmental pollution are controversial, and it is necessary to continue to explore them. Importantly, when studying environmental pollution, biased results are more likely to occur if spatial diffusion effects and externalities are not considered [42,43]. As a new air pollution problem in China, there is less literature on O3 pollution from macro- and meso-perspectives. Secondly, technological innovation plays an important role in atmospheric problems such as ozone pollution, but the effects of technological innovation in China are not clear at present. Thirdly, the scope of the existing literature is either national [23,44] or within a certain region [45,46,47], leading to a lack of comparative analysis of multiple important regions. Therefore, this paper provides contributions in these three aspects to investigate the impact of technological innovation on O3 pollution in key regions of China using spatial econometric models. Additionally, referring to the design of Wang [38], the following hypothesis is proposed in this paper. There may be a relationship between technological innovation and ozone pollution similar to the EKC; i.e., environmental pollution shows an increase and then a decrease with the increase in R&D investment.

## 2. Materials and Methods

This section first introduces the theoretical model of this paper, the STIRPAT model, including the mathematical and theoretical formulations and related assumptions. Then, it introduces the spatial econometric model used in this paper, including its construction and analysis methods.

### 2.1. Theoretical Model and Methodology

#### 2.1.1. The STIRPAT Model

The STIRPAT (Stochastic Impacts by Regression on Population, Affluence, and Technology) model [48], which is based on the IPAT model, is now widely used in technology and environmental economics research. It not only allows for the parameter estimation of coefficients but also allows researchers to decompose and refine each impact factor according to the specific study, allowing other variables to be included in the model to analyze the impact of each factor on environmental stress. It can be illustrated by the equation $I_{\mathrm{it}}{=a\;P}_{\mathrm{it}}{}^{b}{\;A}_{\mathrm{it}}{}^{c}{\;T}_{\mathrm{it}}{}^{d}\;\varepsilon_{\mathrm{it}}$, that is, Stochastic Impacts by Regression on Population, Affluence, Technology, and Error. Here, i and t denote the individual city and time period, respectively; a represents a constant term; and b, c, and d are the estimated parameters. Equation (1) is obtained after taking the natural logarithm of both sides, as follows:(1)$${\mathrm{lnI}}_{\mathrm{it}}{=a+b\;\mathrm{lnP}}_{\mathrm{it}}{+c\;\mathrm{lnA}}_{\mathrm{it}}{+d\;\mathrm{lnT}}_{\mathrm{it}}{+\varepsilon }_{\mathrm{it}}$$
where O_3_ concentration, population density, and GDP per capita represent environmental stress ($I_{\mathrm{it}}$), population ($P_{\mathrm{it}}$), and economic development ($A_{\mathrm{it}}$), respectively. $T_{\mathrm{it}}$, namely, technology innovation, is the focus of the present paper. According to existing studies [22,49,50], the number of patents granted is employed as a proxy for it, and its squared term is introduced to test the hypothesis mentioned previously.

Moreover, the model neglects some other factors influencing O_3_ concentration, such as industrial structure, urbanization, energy use intensity, number of cars, and level of foreign investment utilization, which are often used as explanatory variables for atmospheric pollutants [51,52,53]. Finally, the STIRPAT model is modified to Equation (2).
(2)$$\begin{array}{cc} {\ln\;O}_{3\mathrm{it}} & {=a+d}_{1}{\ln\;\mathrm{tech}}_{\mathrm{it}}{+d}_{2}{\mathrm{lntech}2}_{\mathrm{it}}{+b\;\ln\;\mathrm{pop}}_{\mathrm{it}}{+c\;\ln\;\mathrm{agdp}}_{\mathrm{it}}\\ & {+e\;\ln\;\mathrm{energy}}_{\mathrm{it}}{+f\;\mathrm{ind}}_{\mathrm{it}}{+g\;\ln\;\mathrm{car}}_{\mathrm{it}}\;{+h\;\ln\;\mathrm{fdi}}_{\mathrm{it}}{+\varepsilon }_{\mathrm{it}} \end{array}$$
where $O_{3\mathrm{it}}$ represents the average annual concentration of ozone; ${\mathrm{tech}}_{\mathrm{it}}$ is the number of patents granted; ${\mathrm{pop}}_{\mathrm{it}}$ is the year-end statistics of a city’s number of people per square kilometer; ${\mathrm{agdp}}_{\mathrm{it}}$ is GDP per capita; ${\mathrm{energy}}_{\mathrm{it}}$ measures a city’s energy consumption intensity in terms of electricity consumption per unit of GDP; ${\mathrm{ind}}_{\mathrm{it}}$ is the output value of the secondary industry and accounts for the proportion of total output value; ${\mathrm{car}}_{\mathrm{it}}$ is the number of cars in a city at the end of the year; and ${\mathrm{fdi}}_{\mathrm{it}}$ is the total amount of foreign capital actually utilized in the current year.

#### 2.1.2. Methodology

Due to the role of atmospheric transport, the atmospheric pollutants in each region are not independent of each other and often have a strong spatial correlation. The traditional linear regression analysis method based on the assumption that the observations are independent of each other is not suitable for this analysis, so a spatial regression model is proposed. 

Before the causality analysis, it is first necessary to determine whether the dependent variable is spatially correlated. The global Moran’s I [54] was used to examine the spatial correlation of ozone pollution concentrations, as shown in Equation (3):(3)$$I=\frac{n\;\sum_{i=1}^{n}\;\sum_{j=1}^{n}W_{\mathrm{ij}}\left({\;x}_{i}\;-\bar{\;x}\right)\left(x_{j\;}-\bar{\;x}\right)}{\sum_{i=1}^{n}\;\sum_{j=1}^{n}W_{\mathrm{ij}}\sum_{i=1}^{n}{\left(x_{i\;}-\bar{\;x}\right)}^{2}}$$
where n is the number of spatial cells; $W_{\mathrm{ij}}$ is the spatial weight matrix; $x_{i}$ and $x_{j}$ are the observations at spatial locations i and j, respectively; and $\bar{\;x}$ is the mean value. The range of the global Moran’s I is [–1,1], where I < 0 means negative spatial autocorrelation (i.e., similar observations tend to be spatially dispersed); I=0 means no correlation; and I > 0 means positive spatial autocorrelation (i.e., similar observations tend to be spatially clustered). The global Moran’s I is usually tested for significance using the Z test, as per Equation (4):(4)$$Z\left(I\right)=\frac{I\;-\;E\left(I\right)}{\sqrt{\mathrm{VAR}\left(I\right)}}$$
where E(I) and VAR(I) are the mathematical expectation and variance of I, respectively.

Furthermore, in contrast to the global Moran’s index, which determines the existence of spatial correlation in a region, the local Moran’s I detects the extent and location of internal outliers or agglomerations, as per Equation (5):(5)$$I_{i}=\frac{Z_{i}}{S^{2}}\;\sum_{i\neq j}^{n}w_{\mathrm{ij}\;}Z_{j}$$
where $Z_{i}=y_{i}-\bar{y}$, $Z_{j}=y_{j}-\bar{y}$, $S_{2}=\frac{1}{n}\sum {\left(y_{i}-\bar{y}\right)}^{2}$, $w_{\mathrm{ij}}$ is the spatial weight value, and n is the total number of all regions in the study area, with $Z_{i}$ as the x-axis and $\overset{n}{\underset{i\neq j}{\sum }}w_{\mathrm{ij}}Z_{j}$ as the y-axis. Local Moran’s I scatter plots were divided into four quadrants: high–high or HH (quadrant I)—cities with high levels of ozone pollution are surrounded by neighboring cities with high levels of the variable of concern; low–high or LH (quadrant II)—cities with low levels of ozone pollution are surrounded by neighboring cities with high levels of the variable; low–low or LL (quadrant III)—cities with low levels of ozone pollution are surrounded by neighboring cities with low levels of the variable of concern; and high–low or HL (quadrant IV)—cities with high levels of ozone pollution are surrounded by neighboring cities with low levels of the variable.

Starting from the generalized nested model (Equation (6)), a series of commonly used spatial measurement models can be clearly established [55].
(6)$$\{\begin{array}{c} {Y=\rho }_{1}{\;\mathrm{WY}+\beta \;X+\rho }_{2}\;\mathrm{WX}+\varepsilon \\ {\varepsilon =\eta \;W\varepsilon +\nu ,\;\nu \;\sim \;N\;(0,\;\sigma }^{2}I) \end{array}$$
where Y is the dependent variable matrix; X is the independent variable matrix; and W is the spatial weight matrix, which measures the distance relationship between different regions. Thus, WY and WX represent the interaction effects of the dependent and independent variables, respectively; $\rho_{1}{,\;\beta ,\;\rho }_{2}$, and η are the corresponding regression coefficients, in which $\rho_{1}$ and η are referred to as spatial correlation coefficients; I is a column vector with elements 1; ε is a column vector for the error terms; and thus, Wε is the interaction effect of the error terms.

The common spatial econometric models are the spatial lag model (SLM), spatial error model (SEM), and spatial Durbin model (SDM). In Equation (3), if η = 0, then the model degenerates into SDM; if η = 0 and $\rho_{2}$ = 0, then the model is SLM, which assumes that the variation of the dependent variable in a certain location is affected by the combined change in dependent variables in the surrounding area; if $\rho_{1}{=\rho }_{2}\;$= 0, then the model is SEM, which hypothesizes that the spillover effect comes from the influence of omitted variables of the neighboring region [56]. In addition, in order to observe the difference between short-term and long-term effects, we finally add the lagged dependent variable to build a dynamic spatial model (Equation (7)).
(7)$$\{\begin{array}{c} {Y=\rho }_{0}{\;Y}_{t-1}{+\rho }_{1\;}{\mathrm{WY}+\beta X+\rho }_{2}\;\mathrm{WX}+\varepsilon \\ {\varepsilon =\eta \;W\varepsilon +\nu ,\;\nu \;∼\;N\;(0,\;\sigma }^{2}I) \end{array}$$

Additionally, the spatial weight matrix W needs to be set in advance in Equation (4). In this paper, we constructed two spatial weight matrices. The first is the binary contiguity matrix $W_{1}$ [57], which is made according to queen-based contiguity. When city i and city j have a common boundary or a common node, $w_{\mathrm{ij}}$ = 1; otherwise, $w_{\mathrm{ij}}$ = 0 (Equation (8)). The other is the distance-based weight matrix $W_{2}$ [58], whose element $w_{\mathrm{ij}}$ is the reciprocal of the geographical distance between urban administrative center i and urban administrative center j (Equation (9)).
(8)$$w_{\mathrm{ij}}=\{\begin{array}{c} 1,\;i\;\neq \;j,\;i=1,\;\cdots \;,\;n;\;j=1,\;\cdots \;,\;n\\ 0,\;i=j,\;i=1,\;\cdots \;,\;n;\;j=1,\;\cdots \;,\;n \end{array}$$
(9)$$w_{\mathrm{ij}}=\{\begin{array}{c} 1⁄d\_\mathrm{ij}\;,\;i\;\neq \;j,\;i=1,\;\cdots \;,\;n;\;j=1,\;\cdots \;,\;n\\ 0,\;i=j,\;i=1,\;\cdots \;,\;n;\;j=1,\;\cdots \;,\;n \end{array}$$

As shown in Equation (7), the spatial Durbin model jointly captures the influence of the spatial lag dependent variable and spatial lag explanatory variables, which may lead to the endogeneity problem and violates the classical assumptions of the ordinary least square (OLS) method. Here, we used the maximum likelihood (ML) method to effectively solve the endogeneity problem [59]. Additionally, because of spatial correlation, the change in independent variables in one region will not only directly affect its dependent variable but also affect the dependent variable in the region associated with its existence. Therefore, the coefficients of the variables do not represent the marginal effect (Equation (10)), and instead, more attention should be paid to the “direct effect” (Equation (11)) and “indirect effect” (Equation (12)) [55]. The specific derivation processes are re-written as follows:(10)$$Y={\left({I\;-\;\rho }_{1}\;W\right)}^{-1}\left({\beta \;X+\rho }_{2}\;\mathrm{WX}+\eta \;W\varepsilon +\nu \right)$$
(11)$$\partial y_{i}/\partial x_{\mathrm{ir}}={\left({I\;-\;\rho }_{1}\;W\right)}^{-1}\left(\beta_{r}{\;I+\rho }_{2r}{\;W}_{\mathrm{ii}}\right)$$
(12)$$\partial y_{i}/\partial x_{\mathrm{jr}}\;={\left({I\;-\;\rho }_{1}\;W\right)}^{-1}\left(\beta_{r}{\;I+\rho }_{2r}{\;W}_{\mathrm{ij}}\right),\;i\;\neq \;j$$
where I represents an n × 1 unit matrix, n is the number of cities, ${\left({I\;-\;\rho }_{1}\;W\right)}^{-1}$ stands for the spatial Leontief inverse matrix, $\partial y_{i}/\partial x_{\mathrm{ir}}$ denotes the direct effect, and $\partial y_{i}/\partial x_{\mathrm{jr}}$ refers to the indirect effect. Finally, we used STATA15.0 to estimate the magnitude and signs of the direct and indirect effects in the dynamic model.

### 2.2. Study Area and Data Description

In 2018, China clearly listed the Fenwei Plain (FW) as a key area for air pollution prevention and control alongside Beijing–Tianjin–Hebei (BTH) and the Yangtze River Delta (YRD) in a new update of the air pollution prevention and control action plan. TH, YRD, and FW are important urban agglomerations in the northern, central-eastern, and central-western parts of China, as shown in Figure 1, whose economic volumes are the third, first, and ninth in the country, respectively. However, in these three regions, the number of days with O_3_ as the primary pollutant accounted for 48.2%, 49.5%, and 37.6% of the total exceedance days in 2019, respectively [60]. These three regions include a total of 50 cities: 13 in BTH, 26 in the YRD, and 11 in the FW, as shown in Table 1.

BTH, YRD, and FW were selected as the key study regions, mainly due to their commonalities and differences. From the perspective of commonality, firstly, in 2018, FW replaced the PRD region and became one of the three key air pollution control regions, along with BTH and YRD, due to the prominent problems of ozone pollution and fine particulate matter pollution (Figure 2). Secondly, BTH, YRD, and FW all belong to developed urban agglomerations in China, which are important in terms of economic development, technological level, and population size. From the perspective of differences, firstly, the geographical location and positioning are different. BTH is located in the North China Plain, which is the economic circle of the capital city of China; YRD is located in the lower reaches of the Yangtze River in East China, centered on Shanghai, which is a region with strong openness and high innovation; and FW is located in the middle reaches of the Yellow River in central and western China, with rich mineral resources and high levels of industrial and agricultural production. Next, there are differences in the GDP, size of the population, number of patents, and number of cars in the country, ranking YRD, BTH, and FW in order as follows. In terms of industrial structure, BTH is better than YRD and FW; in terms of energy use, the YRD region creates 23.94% of the GDP with 17% of the energy consumption, while BTH creates only 8.6% of the GDP with 9.6% of the energy consumption, and FW creates 2.93% of the energy consumption and 2.88% of the GDP, as shown in Table 2.

Considering the availability of data, 50 cities in BTH, YRD, and FW, the three major regions of air pollution control in China during 2014–2019, were selected as balanced panel data samples in this paper. The nationwide hourly ozone observations in Chinese cities were obtained from the China National Environmental Monitoring Center (CNEMC) network, and then we obtained the average annual ozone concentration data after taking the arithmetic average. Observations of key independent variables and other control variables are from the *China City Statistical Yearbook* and the statistical yearbooks of various provinces and cities. As for the spatial weight matrix W, the latitude and longitude of each city and the vector map data bounded by administrative areas were obtained from the National Catalogue Service for Geographic Information. Table 3 shows the variables’ descriptive statistics without taking logarithmic values.

## 3. Results

### 3.1. Spatial Autocorrelation Test

In order to avoid the heteroskedasticity problem to some extent, non-ratio variables were treated as natural logarithms when performing regression. Table 4 lists the global Moran’s I of lnO_3_ for three regions each year and the calculated average values, illustrating that most Moran’s I statistic values are greater than zero and statistically significant at the 10% level. It is found that ozone pollution in the three regions of BTH, YRD, and FW is indeed spatially correlated, which indicates that we need to take this spillover effect into extra consideration.

The LISA map for ozone pollution concentrations in 2019 (Figure 3) shows the spatial clustering of O_3_ pollution among cities within the BTH, YRD, and FW regions, indicating that there is a positive spatial correlation in ozone pollution concentration in the three regions, with a high–high and low–low clustering trend. We can see that, in 2019, ozone pollution within the BTH region was highly correlated in Shijiazhuang and Hengshui and lowly correlated in Tangshan, Qinhuangdao, and Chengde. Ozone pollution within the YRD region was highly correlated in Suzhou, Nantong, Yancheng, Yangzhou, and Taizhou and lowly correlated in Hefei, Wuhu, Tongling, Anqing, and Chizhou. Ozone pollution within the FW region was low–low correlated in Jinzhong and Luliang and high–low correlated around Luoyang. This indicates that ozone pollution concentration presents obvious spatial agglomeration. Moreover, the degree of spatial dependence has a tendency to increase.

### 3.2. Estimation Results

#### 3.2.1. Selection of the Specific Model

To determine which spatial econometric model is more appropriate for estimation, first, we conducted conventional OLS estimations and performed the corresponding (robust) LM-lag and LM-error tests for the two spatial econometric estimators, and then we conducted SDM directly and performed the LR and Wald tests to determine whether SDM could be converted to SLM or SEM. Table 5 reports the results of these tests. The values of LM-lag are more significant than those of LM-error, and both the values of Robust LM-lag and Robust LM-error are significant at the 5% level, indicating that the SLM model should be selected in BTH. In YRD, the values of Robust LM-lag are more significant than those of Robust LM-error in the case that both LM-lag and LM-error are significant at the 1% level, so SLM is selected. For FW, SLM is also selected. In addition, the hypothesis that SDM should be converted to SLM or SEM is rejected in the BTH and FW regions; in YRD, the results of the LR test and the Wald test are opposite, and thus, increased attention should be paid to the SDM results. According to the results of the Hausman test, fixed effects should be used in all three regions, and the last LR test indicates that none of the three regions can be simplified to be time-fixed or individual-fixed, so all use double-fixed effects.

#### 3.2.2. Regression Results

Table 6 reports the results of SDM. It is noteworthy that the autocorrelation parameter “rho” for O_3_ concentrations is statistically significant at the 5% level in the three regions, indicating that there are spatial dependences in the sample. Meanwhile, the time-lagged dependent variable (L.O_3_) is significantly and positively associated with O_3_ pollution concentration in BTH and YRD, but not in FW, which is not statistically significant. The current period of O_3_ pollution is indeed affected by the previous period; thus, the problem of controlling it is a long-term and arduous task. It is also very important that the marginal effects of these independent variables on the dependent variable cannot be reflected by the coefficients of SDM [55], and so Table 6 reports the “direct effect”, “indirect effect”, and “total effect” of the variables. Here, a “direct effect” indicates the impact of a city’s independent variables on its O_3_ concentration, and an “indirect effect” presents the impact of the independent variables of neighboring cities on a city’s O_3_ concentration. It can be seen that the “direct effect” of variables is different from the corresponding coefficient values in Table 6, because spatial dependence is considered in the spatial econometric models, and there is a feedback mechanism for the relationship between variables.

#### 3.2.3. Direct Effect and Indirect Effect

As for the key independent variable in this paper, it can be seen from the “total effect” that the increase in technological innovation (lntech) can significantly decrease the O_3_ pollution concentration, and its squared term (lntech2) carries a positive sign in the BTH region. Therefore, the results of lntech and lntech2 indicate a U-shaped relationship between technological innovation and O_3_ pollution. In contrast, more advanced technological innovation aggravates the O_3_ concentration in the YRD and FW regions, and their squared terms are negative, illustrating that O_3_ pollution shows an inverted U-shaped transition when accompanied by technological innovation. However, the former is significant at the 1% level, and FW is not. 

From the “direct” and “indirect” effects (Table 7), in the short term, the direct effects of technological innovation on O_3_ pollution in the BTH region are significantly negative (−0.97) at the 1% level, and the indirect effects are positive (0.135) but not significant. In the YRD region, the direct effect and indirect effect of technological innovation are positive (0.315) and negative (−2.491), respectively, significant at the 10% level, and the absolute value of the indirect effect is much greater than that of the direct effect. Finally, the technological innovation of FW has no significant direct or indirect impact on O_3_ pollution reduction, which eventually leads to a positive and insignificant total effect (0.403) of technological innovation. In the long run, only in the BTH region, the direct effect of technological innovation is significantly negative (−2.463) at the 1% level. None of the technological spillover effects in the three regions are significant, indicating that technological innovation has no long-term effect.

### 3.3. Model Validation

The endogeneity problem in this paper focuses on the possible inverse causality between O_3_ pollution and technological innovation: that is, technological innovation may inhibit or exacerbate ozone pollution conditions, but at the same time, ozone pollution also causes changes in the level of technological innovation. In addition to the dynamic spatial panel model (with a one-period lag of O_3_ pollution added as the dependent variable) used in the previous paper to try to address the endogeneity issue, the key explanatory variable L.tech with a one-period lag was used here as an instrumental variable (IV) to test the endogeneity issue. As can be seen from the results in Table 8, the positive and negative signs and significance of the key explanatory variables are basically consistent with the previous paper, and the estimation results of the coefficients of other variables also remain generally consistent, indicating that the model setting in this paper is reasonable.

Taking into account the conclusion’s robustness, we chose to replace the spatial weight matrix to conduct a robustness test. Here, the inverse distance matrix was employed to measure the coefficient value. The results in Table 9 prove that the coefficient values of the key dependent variable and its squared term are both significant and mainly consistent with the original results, regardless of which spatial weight matrix is used. Additionally, most of the other control variables are close to the results of the binary contiguity matrix.

## 4. Discussion

This research investigated the spatial impact of technological innovation on ozone pollution and its regional differences. Overall, current technological innovations in China, represented by the three major regions, are not very effective in mitigating ozone pollution, which is consistent with the findings of Chen [23] for a full sample of countries. From the perspective of heterogeneity, we focus on the differences among the three major regions.

### 4.1. Technology Innovation

First, in the BTH region, the total effect of high-level technological innovation is a relatively significant mitigation of ozone pollution, which is consistent with the findings of Chen [23] and Churchill [51]. For high-technology-level countries, developed regions have a strong innovation drive to develop environmentally friendly technologies that can effectively improve regional pollution. On the one hand, in front-end prevention, the BTH region, with a high level of technological innovation, applies clean production technologies and environmental protection products to energy systems and production systems, which can essentially improve the efficiency of enterprises’ resource ecological use and reduce the generation of ozone precursors. The middle-end process uses cutting-edge pollution-monitoring platforms to mitigate ozone precursors and has timely access to pollution information. In the terminal treatment, they help enterprises improve their pollution treatment capacity and development capacity to reduce the emission of haze-causing substances with high-technology R&D capability. On the other hand, technological innovation will also further reduce the production of ozone precursors by optimizing resource allocation and promoting industrial structure upgrading. By leading the flow and concentration of capital, labor, technology, and information from traditional industries to new industries, technology-intensive and low-pollution industries can optimize industrial and resource allocation structures, improving the allocation efficiency and total factor productivity.

Second, in the YRD region, it is noteworthy that its relatively high level of technological innovation has instead exacerbated regional ozone pollution, which is consistent with the findings of Fernandez et al. [61], Ullah [62], and Chen [23]. The direct effect of technological innovation is significantly positive, while the indirect effect is significantly negative, with the final total effect being positive at the 1% significance level. On the one hand, this may be due to the fact that regional demand-driven technological innovation focuses mainly on industrial support and ecological growth, which effectively promotes local economic growth at the expense of environmental quality to some extent. On the other hand, the YRD region contains the largest number of cities and has a large variability in internal development. Although the spillover effect due to the flow of technological innovation factors has a mitigating effect on ozone pollution, this effect is limited.

Finally, in the FW region, it can be seen that the effect of technological innovation on regional ozone pollution is not significantly positive and is small in absolute value. This is consistent with the findings of Chen [23] for low-income countries, Samargandi [36] for Saudi Arabia, and Cheng et al. [34] for OECD. Collectively, the FW region has the lowest level of economic development and innovative technology among the three regions, making it difficult to meet the technology and requirements of the green threshold to effectively improve environmental quality, resulting in a negligible effect.

### 4.2. Other Influencing Factors

Here, we discuss the causes of the large differences in the effects of each control variable in the three regions of BTH, YRD, and FW. Overall, this is largely consistent with the development characteristics and resource endowment differences among the three regions and justifies the study to a certain extent. The specific analysis is as follows.

#### 4.2.1. The BTH Region

In the BTH region, the direct effects of the economic development level and foreign direct investment on ozone pollution are significantly positive, in addition to the technological innovation factor. This indicates that ozone pollution has a positive relationship with the economic level and FDI level, which is consistent with the classical “pollution paradise hypothesis” proposed by Kathuria [63]. BTH’s GDP accounted for 8.6% of the national share, and FDI accounted for 20.56%, ranking second among the three regions. Both ranked second in 2019. In the process of rapid economic development, the massive entry of foreign investment can lead to the concentration of highly polluting enterprises in areas with lower environmental regulations and access barriers, which can generate a large number of ozone pollution precursors. The direct effect of the industrial structure on ozone pollution is significantly negative, which indicates that the industrial structure of the BTH region is to some extent conducive to improving regional ozone pollution. In 2019, the structure of the three major industries in the Beijing–Tianjin–Hebei region was the best among the three regions, with the proportions of primary, secondary, and tertiary output values at 4.5%, 28.7%, and 66.8%, respectively, and the proportion of secondary industry output value as low as 28.7%, which makes an important contribution to the mitigation of ozone pollution in the region. Second, the spillover effects of motor vehicle numbers and industrial structure are significantly positive, which is consistent with Silva [64], Anenberg [65], and Unger [66]. The large amounts of NO_X_ and VOC_S_ emitted from the exhaust of motor vehicles such as cars and diesel vehicles become precursors of ozone pollution, and their casual mobility contributes to air pollution in the surrounding cities. In addition, the transfer of highly polluting industries, industrial undertaking, and factor mobility within the region are also the main reasons that the industrial structure aggravates ozone pollution.

#### 4.2.2. The YRD Region

In the YRD region, the direct effects of the economic development level and the number of motor vehicles on ozone pollution are significantly negative, except for the effect of technological innovation. This indicates that the regional economic development and the number of motor vehicles do not exacerbate ozone pollution but rather have a mitigating effect on it. This may be consistent with the second half of the EKC curve [24], where the ozone pollutants show a decreasing trend with the increasing economic level in the YRD region. The relatively consistent and well-established public green transportation system in the YRD also mitigates ozone pollution to some extent. Other control variables do not have significant effects on ozone pollution in the YRD, which is consistent with the regression results of Chen [23] for the whole study sample. On the one hand, spillover effects from adjacent areas may offset the effects of a local factor on ozone pollution; on the other hand, the YRD includes more cities and intra-regional heterogeneity prevails, and the overall effect of a factor on ozone pollution may not be significant.

#### 4.2.3. The FW Region

In the FW region, first, the direct and spillover effects of energy use intensity on ozone pollution are significantly positive, indicating that regional energy use is the main factor that exacerbates local ozone pollution concentrations. This is consistent with Radmehr [67] and Adewuyi [68]. In 2019, energy consumption in the FW accounted for 2.93% of the national share, with coal accounting for 80% of the total, the highest among the three regions. The fossil-energy-based energy structure and higher energy use intensity in particular have a significant impact on environmental pollution and are the main cause of ozone pollution. Second, the direct effect of the number of motor vehicles is significantly positive, with the number of motor vehicles in the FW reaching 9,298,000 in 2019, the main source of mobile emissions causing ozone pollution. This, coupled with the low topography of the FW itself—it is in the valley of the Fen and Wei rivers—is not conducive to the diffusion of pollution sources, making pollution worse. In addition, both the direct and indirect effects of foreign direct investment are significantly negative, which is consistent with Gunnar [69]. This indicates that foreign investment in the FW is mainly environmentally friendly and promotes the regional environmental technology level through the “demonstration effect”, “spillover effect”, and “competition effect”, which has a mitigating effect on ozone pollution. The direct effect of population density on ozone pollution is significantly negative, which may be mainly due to the relatively low population density in the FW region, with the resident population accounting for 3.68% of the national population in 2019, the lowest among the three regions.

## 5. Conclusions

This paper explores the impact of technological innovation on O_3_ pollution in three important regions of China (Beijing–Tianjin–Hebei, the Yangtze River Delta, and the Fenwei Plain) over the period 2014–2019 through spatial econometric models. The main contribution is the analysis of the spillover effects of technological innovation on O_3_ pollution and their heterogeneity. Generally, there is an obvious spatial correlation of O3 pollution, and technological innovations in China are still not effective in curbing ozone pollution. Secondly, the empirical results first prove that O_3_ pollution and technological innovation are spatially correlated among cities of a region, and the spillover effect is important in understanding the relationship between technological innovation and O_3_ pollution. Thirdly, technological innovation in BTH not only significantly reduces its O_3_ pollution but also helps reduce O_3_ pollution in neighboring cities in the long term; the indirect effect of this in YRD is significantly negative, but the total effect is still a positive contribution to ozone pollution. Technological innovation in FW does not significantly promote O_3_ pollution in its own cities and/or neighboring cities. Furthermore, the coefficients of the control variables show that the main influencing factors of O_3_ pollution vary from region to region. This is in line with the differences in endowments and development characteristics of each region. Based on this paper’s main conclusions, some relevant policy implications are suggested, as follows:The positive superposition of O_3_ pollution in time and space dimensions indicates that it is urgent to carry out regional control and joint prevention efforts to limit regional O_3_ pollution. In particular, key regions should actively carry out intra- and inter-regional synergistic cooperation and implement joint actions in key pollution source monitoring, mobile monitoring, legislative enforcement, quantitative standards, etc., to strictly implement O_3_ pollution regulation and pollution management.The direct and indirect effects of technological innovation on ozone pollution vary considerably between regions. In particular, in the YRD and FW regions, on the one hand, they should promote green environmental protection technologies, accelerate the elimination of energy-consuming and polluting production technologies, and encourage international corporations to help cities with lower technological innovation. On the other hand, they should pay more attention to improving O_3_ pollution control technologies and equipment, including controlling the sources of pollution and end-of-treatment.As for other factors on ozone pollution, the BTH region should strengthen the management of motor vehicles and promote a green transportation system throughout the whole area while strengthening the upgrading of industrial structure, especially in the cities of Hebei Province. The YRD region should improve its energy use efficiency and reduce its energy use intensity. The FW region, because of its abundant energy resources, mostly coal, has an industrial structure dominated by secondary industries and high energy use intensity, which has exacerbated the degree of ozone pollution. Therefore, efforts should focus on the upgrading of industrial structure and energy use to promote the formation of green industries and energy use.

## Figures and Tables

**Figure 1 ijerph-19-07743-f001:**
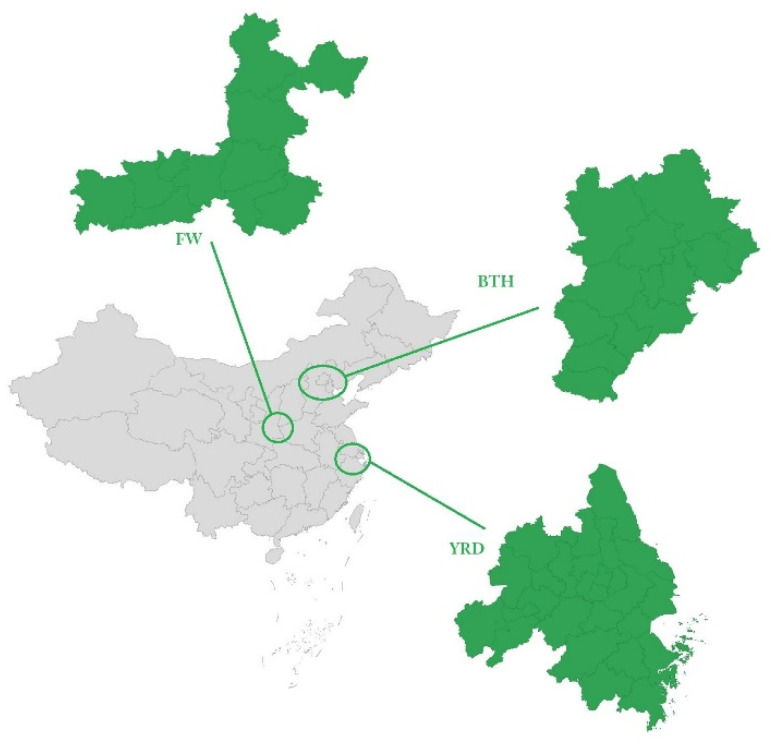
Location of BTH, YRD, and FW in China.

**Figure 2 ijerph-19-07743-f002:**
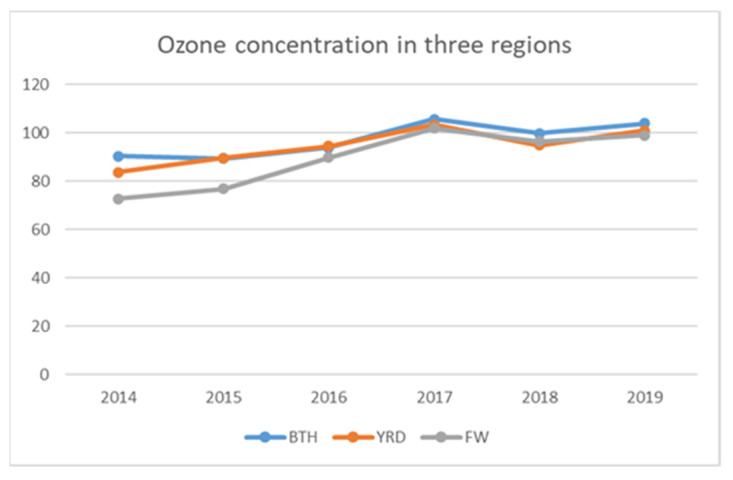
Ozone concentration trends in BTH, YRD, and FW from 2014 to 2019 (μg m^−3^).

**Figure 3 ijerph-19-07743-f003:**
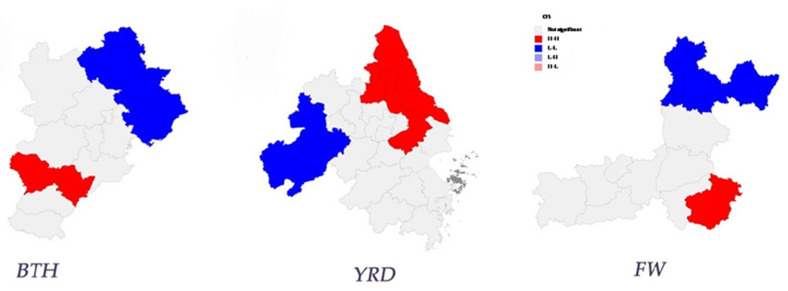
LISA map of lnO_3_ concentration in 2019.

**Table 1 ijerph-19-07743-t001:** Cities included in the three regions.

Region	City Name
BTH	Beijing, Tianjin, Shijiazhuang, Tangshan, Qinhuangdao, Handan, Xingtai, Baoding, Zhangjiakou, Chengde, Cangzhou, Langfang, Hengshui
YRD	Shanghai, Nanjing, Wuxi, Changzhou, Suzhou, Nantong, Yancheng, Yangzhou, Zhenjiang, Taizhou ^1^, Hangzhou, Ningbo, Jiaxing, Huzhou, Shaoxing, Jinhua, Zhoushan, Taizhou ^1^, Hefei, Wuhu, Maanshan, Tongling, Anqing, Chuzhou, Chizhou, Xuancheng
FW	Jinzhong, Yuncheng, Linfen, Lvliang, Luoyang, Sanmenxia, Xi’an, Tongchuan, Baoji, Xianyang, Weinan

^1^ Note: Two different cities: the former belongs to Jiangsu Province, China, and the latter belongs to Zhejiang Province, China.

**Table 2 ijerph-19-07743-t002:** Comparison of the differences between BTH, YRD, and FW regions in 2019.

Region	Position	GDP Share of the Country (%)	Population Share of the Country (%)	Number of Patents Granted (Million)	Ratio of the Three Industries	Number of Cars (Million)	FDI Share of the Country (%)	Energy Consumption Share of the Country (%)
BTH	Located in the North China Plain	8.6	8.1	24.55	4.5: 28.7: 66.8	2707.81	20.56	9.6, of which coal accounted for 67.9
YRD	Located in the lower reaches of the Yangtze River in East China	23.94	16.7	67.9	3.97: 40.66: 55.37	4131.42	49.86	17%, of which coal accounted for 55.4%
FW	Located in west-central China, in the middle reaches of the Yellow River	2.88	3.68	4.59	8.11: 40.36: 51.54	929.81	8.07	2.93%, of which coal accounted for 80%

**Table 3 ijerph-19-07743-t003:** Descriptive statistics of variables.

Region	Variable	Unit	Mean	Std. dev	Min	Max
BTH	O_3_	${\mu g\;m}^{-3}$	97.1766	10.4500	65	113.5
	tech	--	14,308.45	29,244.58	300	13,1716
	energy	${\mathrm{kwh}\;\mathrm{yuan}}^{-1}$	0.0980	0.1024	0.0152	0.6761
	agdp	${\mathrm{yuan}\;\mathrm{person}}^{-1}$	55,960.36	31,666.73	22,758	164,220
	car	--	181.8881	137.2016	51.56	636.5
	ind	%	41.1321	10.7247	8.5	62.1
	pop	${\mathrm{person}\;\mathrm{km}}^{-2}$	606.7331	313.0762	61.9616	1324
	fdi	yuan	262,370.4	519,982.3	1047	2,432,909
YRD	O_3_	${\mu g\;m}^{-3}$	94.4653	15.3034	36	115
	tech	--	20,903.1	19,415.71	1040	100,587
	energy	${\mathrm{kwh}\;\mathrm{yuan}}^{-1}$	0.0538	0.0250	0.0124	0.1153
	agdp	${\mathrm{yuan}\;\mathrm{person}}^{-1}$	92,849.81	35,715.66	28,808	180,044
	car	--	130.126	94.9818	8.73	442.55
	ind	%	47.2482	6.6954	26.99	68.27
	pop	${\mathrm{person}\;\mathrm{km}}^{-2}$	790.3617	677.264	171.8062	3830
	fdi	yuan	258,718.2	353,974.1	7792	1,904,800
FW	O_3_	${\mu g\;m}^{-3}$	89.4681	15.0859	58.75	118.6667
	tech	--	3458.439	8278.139	117	38,279
	energy	${\mathrm{kwh}\;\mathrm{yuan}}^{-1}$	0.0673	0.0536	0.0069	0.2247
	agdp	${\mathrm{yuan}\;\mathrm{person}}^{-1}$	44,500.66	16,658.41	22,304	92,256
	car	--	66.1168	69.0366	6.9349	343.0559
	ind	%	47.3930	9.2123	32.84	70.04
	pop	${\mathrm{person}\;\mathrm{km}}^{-2}$	348.3166	191.5584	180.7995	863.2962
	fdi	yuan	87,823.59	156,464.5	1	665,666

**Table 4 ijerph-19-07743-t004:** Global Moran’s I of lnO_3_.

Year	BTH	YRD	FW
2014	−0.097	0.513 ***	0.209
2015	−0.091	0.586 ***	0.469 ***
2016	−0.100	0.285 ***	0.235 **
2017	0.191	0.038	−0.021
2018	0.480 ***	0.106	−0.002
2019	0.440 ***	0.135	0.425 **

Note: *** and ** represent significance levels of 1% and 5% respectively.

**Table 5 ijerph-19-07743-t005:** Results of LM, LR, and Wald tests.

Test		BTH	YRD	FW
LM-lag		3.386 **	28.027 ***	15.685 ***
Robust LM-lag		11.119 ***	11.704 ***	2.843 *
LM-error		0.105	16.534 ***	14.143 ***
Robust LM-error		7.838 ***	0.211	1.301
Hausman test		275.73 ***	70.98 ***	5.00
LR test	SDM/SLM	chi2 = 27.17 ***	chi2 = 11.36	chi2 = 21.02 **
	SDM/SEM	chi2 = 29.61 ***	chi2 = 25.98 ***	chi2 = 21.01 **
Wald test	SDM/SLM	chi2 = 50.77 ***	chi2 = 40.06 ***	chi2 = 43.15 ***
	SDM/SEM	chi2 = 70.61 ***	chi2 = 49.15 ***	chi2 = 35.34 ***
LR test	ind/both	chi2 = 17.17 *	chi2 = 19.32 **	chi2 = 40.89 ***
	time/both	chi2 = 44.82 ***	chi2 = 104.80 ***	chi2 = 41.62 ***

Note: ***, **, and * represent significance levels of 1%, 5%, and 10%, respectively.

**Table 6 ijerph-19-07743-t006:** Estimation results of dynamic SDM for the full sample.

Variable	BTH	YRD	FW
L.lnO_3_	0.593 ***	0.514 ***	0.217
	(4.565)	(5.996)	(1.366)
lntech	−1.001 ***	0.525 ***	0.041
	(−5.159)	(2.660)	(0.304)
lntech2	0.057 ***	−0.020 **	−0.003
	(4.900)	(−2.230)	(−0.199)
lnenergy	0.016	0.007	0.054 **
	(1.279)	(0.441)	(2.189)
lnagdp	0.255 *	−0.207 *	0.104
	(1.882)	(−1.795)	(0.342)
lncar	0.245	−0.114 *	0.783 ***
	(1.544)	(−1.904)	(2.831)
ind	−0.008 **	0.003	−0.004
	(−2.491)	(0.854)	(−0.026)
lnpop	0.002	0.534	−1.831 **
	(0.049)	(1.575)	(−2.559)
lnfdi	0.032 ***	−0.016	−0.027 ***
	(2.730)	(−0.658)	(−5.231)
W.lntech	0.394	−1.568 **	0.186
	(0.678)	(−2.062)	(0.393)
W.lntech2	−0.032	0.069 **	−0.017
	(−0.955)	(2.264)	(−0.521)
W.lnenergy	−0.002	0.010	0.116 **
	(−0.075)	(0.330)	(2.328)
W.lnagdp	−0.035	−0.079	−1.340 ***
	(−0.320)	(−0.409)	(−2.756)
W.lncar	0.527 ***	0.058	−0.148
	(4.043)	(0.461)	(−0.361)
W.ind	0.015 ***	−0.006	0.786 **
	(5.021)	(−0.860)	(2.484)
W.lnpop	0.168	−1.114	−0.471
	(0.979)	(−1.065)	(−0.473)
W.lnfdi	−0.016	0.019	−0.036 **
	(−0.712)	(0.281)	(−2.433)
rho	0.303 **	0.491 ***	0.259 **
	(2.543)	(8.418)	(2.572)
sigma2_e	0.002 ***	0.006 ***	0.003 ***
	(6.323)	(2.756)	(4.908)
R^2^	0.7645	0.6593	0.8406

Note: ***, **, and * represent significance levels of 1%, 5%, and 10%, respectively.

**Table 7 ijerph-19-07743-t007:** Marginal effects of dynamic SDM for the full sample.

	Variable	BTH	YRD	FW
SR_Direct	lntech	−0.970 ***	0.315 *	0.074
		(−5.096)	(1.812)	(0.577)
	lntech2	0.054 ***	−0.011	−0.003
		(2.612)	(−1.347)	(−0.108)
	lnenergy	0.017	0.009	0.065 **
		(1.008)	(0.503)	(2.030)
	lnagdp	0.273 *	−0.238 **	−0.020
		(1.930)	(−2.128)	(−0.058)
	lncar	0.308 *	−0.114 **	0.815 ***
		(1.828)	(−2.028)	(3.002)
	ind	−0.007	0.003	0.073
		(−1.119)	(0.722)	(0.496)
	lnpop	0.020	0.409	−1.991 ***
		(0.447)	(1.087)	(−2.927)
	lnfdi	0.032	−0.013	−0.031 **
		(1.563)	(−0.404)	(−2.002)
SR_Indirect	lntech	0.135	−2.491 *	0.329
		(0.173)	(−1.897)	(0.536)
	lntech2	−0.021	0.112 **	−0.030
		(−0.455)	(2.148)	(−0.670)
	lnenergy	0.002	0.022	0.157 **
		(0.075)	(0.364)	(2.574)
	lnagdp	0.087	−0.347	−1.691 **
		(0.606)	(−0.971)	(−2.344)
	lncar	0.828 ***	−0.019	0.097
		(3.456)	(−0.089)	(0.185)
	ind	0.017 **	−0.007	1.013 **
		(2.476)	(−0.584)	(2.479)
	lnpop	0.228	−1.450	−1.308
		(1.005)	(−0.812)	(−0.929)
	lnfdi	−0.007	0.024	−0.058 **
		(−0.223)	(0.185)	(−2.207)
LR_Direct	lntech	−2.463 ***	−0.847	0.116
		(−4.985)	(−0.016)	(0.642)
	lntech2	0.140 ***	0.050	−0.006
		(2.632)	(0.023)	(−0.154)
	lnenergy	0.041	−0.003	0.089 **
		(0.957)	(−0.002)	(2.133)
	lnagdp	0.659 *	−0.804	−0.087
		(1.892)	(−0.049)	(−0.179)
	lncar	0.602	−0.392	1.061 ***
		(1.450)	(−0.094)	(2.990)
	ind	−0.020	0.005	0.132
		(−1.236)	(0.024)	(0.658)
	lnpop	−0.008	−0.211	−2.634 ***
		(−0.072)	(−0.005)	(−2.938)
	lnfdi	0.081	0.033	−0.042 **
		(1.587)	(0.010)	(−2.011)
LR_Indirect	lntech	1.047	−3.021	0.516
		(0.679)	(−0.006)	(0.549)
	lntech2	−0.083	0.324	−0.047
		(−0.907)	(0.014)	(−0.663)
	lnenergy	−0.010	0.057	0.226 ***
		(−0.161)	(0.002)	(2.658)
	lnagdp	−0.025	−6.107	−2.385 **
		(−0.073)	(−0.033)	(−2.153)
	lncar	1.330 **	1.655	0.276
		(2.210)	(0.027)	(0.351)
	ind	0.039 **	−0.168	1.445 **
		(2.232)	(−0.086)	(2.305)
	lnpop	0.406	−31.882	−2.197
		(0.944)	(−0.035)	(−0.971)
	lnfdi	−0.039	−1.222	−0.086 **
		(−0.600)	(−0.017)	(−1.979)

Note: ***, **, and * represent significance levels of 1%, 5%, and 10%, respectively.

**Table 8 ijerph-19-07743-t008:** Results of endogeneity test.

Variable	BTH	YRD	FW
L.lnO_3_	0.549 ***	0.453 ***	0.093
	(5.825)	(13.675)	(0.601)
lnltech	−0.531 ***	0.548 **	0.053
	(−2.809)	(2.531)	(0.360)
lnltech2	0.038 ***	−0.023 **	−0.003
	(3.129)	(−2.029)	(−0.359)
lnenergy	0.019	0.015	0.052 **
	(1.499)	(0.612)	(2.223)
lnagdp	0.100	−0.188 **	0.268 *
	(1.000)	(−1.990)	(1.810)
lncar	−0.055	−0.031	0.654 ***
	(−0.343)	(−0.431)	(2.788)
ind	−0.007 **	0.002	0.001
	(−2.073)	(0.556)	(0.285)
lnpop	0.031	0.452	−1.646 **
	(0.988)	(1.083)	(−2.359)
lnfdi	0.011	0.020	−0.023 ***
	(0.764)	(0.515)	(−4.023)
rho	0.300 ***	0.404 ***	0.037
	(2.620)	(4.044)	(0.239)
sigma2_e	0.003 ***	0.005 ***	0.003 ***
	(7.249)	(2.957)	(6.288)

Note: ***, **, and * represent significance levels of 1%, 5%, and 10%, respectively.

**Table 9 ijerph-19-07743-t009:** Results of robustness test.

Variable	BTH	YRD	FW
L.lnO_3_	0.540 ***	0.603 ***	0.187
	(6.423)	(15.307)	(1.407)
lntech	−0.899 ***	0.229 **	0.115
	(−5.941)	(2.498)	(0.490)
lntech2	0.049 ***	−0.006 *	−0.012
	(4.728)	(−1.827)	(−0.795)
lnenergy	0.006	−0.008	0.039
	(0.648)	(−0.459)	(1.382)
lnagdp	0.246 *	−0.316 ***	0.015
	(1.820)	(−4.065)	(0.063)
lncar	0.491***	−0.090	0.736 *
	(3.449)	(−1.389)	(1.848)
ind	−0.003	0.005 *	−0.003
	(−1.254)	(1.735)	(−0.799)
lnpop	0.107 **	0.521	−1.781 ***
	(2.503)	(1.244)	(−3.136)
lnfdi	0.047 ***	−0.032 *	−0.036 ***
	(3.762)	(−1.665)	(−4.355)

Note: ***, **, and * represent significance levels of 1%, 5%, and 10%, respectively.

## Data Availability

The data presented in this study are openly available in the China National Environmental Monitoring Center (CNEMC) network (http://www.cnemc.cn/en/ (accessed on 20 September 2021)), China City Statistical Yearbook and the statistical yearbooks of related provinces and cities, and National Catalogue Service for Geographic Information (https://www.webmap.cn/mapDataAction.do?method=forw&resType=5&secClass (accessed on 4 October 2021)), where the China City Statistical Yearbook and the statistical yearbooks of various provinces’ and cities’ data sources are specified as follows. China City Statistical Yearbook: https://data.cnki.net/trade/Yearbook/Single/N2020050229?zcode=Z011 (accessed on 23 September 2021). Bei Jing Statistical Yearbook: http://nj.tjj.beijing.gov.cn/nj/main/2021-tjnj/zk/indexch.htm (accessed on 23 September 2021). Tian Jin Statistical Yearbook: http://stats.tj.gov.cn/tjsj_52032/tjnj/ (accessed on 24 September 2021). He Bei Statistical Yearbook: http://tjj.hebei.gov.cn/hetj/tjsj/jjnj/ (accessed on 24 September 2021). Shang Hai Statistical Yearbook: https://tjj.sh.gov.cn/tjnj/index.html (accessed on 25 September 2021). Jiang Su Statistical Yearbook: http://stats.jiangsu.gov.cn/2021/indexc.htm (accessed on 25 September 2021). Zhe Jiang Statistical Yearbook: https://tjj.zj.gov.cn/col/col1525563/index.html (accessed on 28 September 2021). An Hui Statistical Yearbook: http://tjj.ah.gov.cn/ssah/qwfbjd/tjnj/index.html (accessed on 28 September 2021). Shaanxi Statistical Yearbook: http://tjj.shaanxi.gov.cn/tjsj/ndsj/tjnj/ (accessed on 29 September 2021). Shanxi Statistical Yearbook: http://www.shanxi.gov.cn/sj/tjnj/ (accessed on 29 September 2021). He Nan Statistical Yearbook: https://tjj.henan.gov.cn/tjfw/tjcbw/tjnj/ (accessed on 30 September 2021).

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
