# Peer review of "Does Technological Innovation Curb O3 Pollution? Evidence from Three Major Regions in China"

_ijerph, 2022, doi:10.3390/ijerph19137743_

Round 1

Reviewer 1 Report

The topic on O3 pollution is important and the results in this paper are reliable. The paper is suitable for publication after minor modification.

 1) The authors are asked to pay special attention to English writing. The author's unit was wrongly written as Shaanxi Normal University, and should be Shanxi Normal University.

2) This paper studies the effects of technological innovation on O3 pollution. Besides the Durbin model, the authors should introduce some other advanced technologies to study the inter-regional differences in three major regions in China, for examples, the two-scale fractal model.

3) The main factors affecting O3 pollution should be discussed. See the following article for a detailed discussion:

Avnery, S; Mauzerall, DL; (...); Horowitz, LW. Global crop yield reductions due to surface ozone exposure: 2. Year 2030 potential crop production losses and economic damage under two scenarios of O-3 pollution,  ATMOSPHERIC ENVIRONMENT 45 (13)(2011) , pp.2297-2309

Round 2

Reviewer 2 Report

Thank for addressing all the suggestions. You did a nice job. I believe that the quality of the paper is now high enough to be published